# High-Energy X-Ray Compton Scattering Imaging of 18650-Type Lithium-Ion Battery Cell

**Kosuke Suzuki** [1,*], **Ari-Pekka Honkanen** [2], **Naruki Tsuji** [3], **Kirsi Jalkanen** [4], **Jari Koskinen** [4], **Hideyuki Morimoto** [1], **Daisuke Hiramoto** [1], **Ayumu Terasaka** [1], **Hasnain Hafiz** [5,6], **Yoshiharu Sakurai** [3], **Mika Kanninen** [4], **Simo Huotari** [2], **Arun Bansil** [6], **Hiroshi Sakurai** [1] and **Bernardo Barbiellini** [6,7]

1   Faculty of Science and Technology, Gunma University, Kiryu, Gunma 376-8515, Japan
2   Department of Physics, University of Helsinki, P.O. Box 64, FI-00014 Helsinki, Finland
3   Japan Synchrotron Radiation Research Institute, SPring-8, Sayo, Hyogo 679-5198, Japan
4   Akkurate Oy, Kaarikatu 8b, 20760 Kaarina, Finland
5   Department of Mechanical Engineering, Carnegie Mellon University, Pittsburgh, PA 15213, USA
6   Department of Physics, Northeastern University, Boston, MA 02115, USA
7   Department of Physics, School of Engineering Science, Lappeenranta University of Technology (LUT University), FI-53851 Lappeenranta, Finland
*   Correspondence: kosuzuki@gunma-u.ac.jp; Tel.: +81-277-30-1714

**Abstract:** High-energy synchrotron X-ray Compton scattering imaging was applied to a commercial 18650-type cell, which is a cylindrical lithium-ion battery in wide current use. By measuring the Compton scattering X-ray energy spectrum non-destructively, the lithiation state in both fresh and aged cells was obtained from two different regions of the cell, one near the outer casing and the other near the center of the cell. Our technique has the advantage that it can reveal the lithiation state with a micron-scale spatial resolution even in large cells. The present method enables us to monitor the operation of large-scale cells and can thus accelerate the development of advanced lithium-ion batteries.

**Keywords:** 18650 lithium-ion battery; lithiation state; Compton scattering; nondestructive measurements

## 1. Introduction

Compton scattering imaging using high-energy synchrotron X-rays is a unique technique for characterizing the local lithiation state in a lithium-ion battery. A great advantage of this technique is that high-energy X-ray photons with energies over 100 keV can easily penetrate closed electrochemical cells. For example, for 100-keV X-rays, the absorption coefficient for stainless steel (0.3615 cm$^{-1}$) is about two orders of magnitude smaller than that for 20 keV (24.68 cm$^{-1}$)[1]. Therefore, non-destructive measurements become possible not only for test cells but also for commercially available lithium-ion batteries. Moreover, the Compton profile, $J(p_z)$, which is measured by this technique, can be related to the ground-state electron momentum density $\rho(\mathbf{p})$ via the following double integral [2]:

$$J(p_z) = \iint \rho(\mathbf{p})dp_x dp_y \tag{1}$$

where $\mathbf{p} = (p_x, p_y, p_z)$ is the electron momentum, and the momentum density can be expressed as [3,4]

$$\rho(\mathbf{p}) = \sum_j n_j \left| \int \Psi_j(\mathbf{r}) \exp(-i\mathbf{p} \cdot \mathbf{r}) d\mathbf{r} \right|^2 \tag{2}$$

where $\Psi_j(\mathbf{r})$ is the wavefunction of the electron in state $j$, and $n_j$ is the corresponding occupation number.

Since the Compton profile is different for various orbitals, the specific electron orbitals involved in the reduction-oxidation (redox) reaction at the electrodes can be identified by combining experimental Compton profiles with parallel first-principles simulations. In this way, the redox orbitals in $LiMn_2O_4$, $LiCoO_2$, and $LiFePO_4$ cathode materials have been revealed and visualized [5–7]. In fact, by analyzing Compton line-shapes, a quantitative analysis of lithium concentration can be performed, and we developed a technique for direct imaging of the lithiation state of a commercial lithium-ion battery based on Compton scattering. We demonstrated the efficacy of this technique by applying it to commercial lithium battery CR2032 under discharge, where the Compton scattered X-ray intensities were shown to reveal the migration of lithium ions into the positive electrode and clarify the structural changes resulting from the volume expansion of the electrode [8]. Lithium concentration can be determined quantitatively from the shape of the Compton profile through a line-shape parameter analysis, known as *S*-parameter analysis [9], and we successfully observed lithium compositions in the commercial lithium rechargeable battery VL2020 at the positive and negative electrodes while it is cycled [10]. The *S*-parameter analysis has also been used to show the dependency of the charge–discharge rate on the lithiation state distribution [11]. All aforementioned studies, however, involved coin-type cells.

In this study, we applied the Compton scattering imaging technique for the first time to a large commercial 18650-type cylindrical cell with 18-mm diameter and 65-mm height. Our aim was to non-destructively and directly observe lithiation states in local regions of the cell. Many non-destructive X-ray methods based on diffraction and absorption techniques have been used with custom-made laboratory cells using low-energy X-rays with the measurement target limited to either the anode or the cathode [12–14]. An operando X-ray computed tomography (CT) study of a commercial battery has also been reported [15]. However, none of these techniques directly monitor lithium ions. Although neutron diffraction is a promising non-destructive probe, it fails to observe local regions in the cell [16,17]. The present Compton scattering based approach thus offers advantages over other methods in probing the structure of commercial large-scale cells non-destructively.

## 2. Experimental

### 2.1. Commercial 18650-Type Lithium-Ion Cells

The 18650-type lithium-ion cell (model MH1), made by LG Chem, Ltd. (Seoul, South Korea), was used as illustrated in Figure 1a. This cell includes a graphite anode (0.19-mm total thickness including 2-sided electrode coating on 0.015-mm thick Cu current collector foil), a $Li(Ni_{1-x-y},Mn_x,Co_y)O_2$ (NMC) cathode (0.15-mm total thickness including 2-sided electrode coating on 0.025-mm thick Al current collector foil), and a polymer film separator of thickness <0.015 mm. The voltage and nominal capacity were 3.67 V and 3200 mAh, respectively. In order to study the degradation of the cell during cycling, fresh and aged cells were investigated. The aged cell was prepared by cycling it 1395 times at 45 °C. During the cycling process, the cell was charged with a constant current (CC) of 1600 mA (corresponding to 0.5 C rate) until the 4.2-V cut-off voltage was reached. Afterwards, the charging was continued in constant voltage (CV) mode at 4.2 V until the current decreased to 320 mA (0.1 C). Discharge was performed with a constant current of 3200 mA (corresponding to 1 C rate) until the 2.8-V cut-off voltage was reached. A reference cycle was performed at 250 cycle intervals with the same charging procedure but with a constant current of 640 mA (0.2 C) and a cut-off voltage of 3.0 V for the discharge. The discharge capacity of the 18650-cell decreased from 3113.1 mAh to 2215.2 mAh after the charge-discharge cycle was repeated 1395 times. This means that the capacity faded about 29%. We also confirmed a notable increase in the resistance of the electrode through the appearance of a large semi-circle in the impedance spectrum of the aged cell in comparison to the fresh cell. Results of the cycling performance of the aged cell and the impedance spectra of fresh and aged cells are given in the Supplementary Materials.

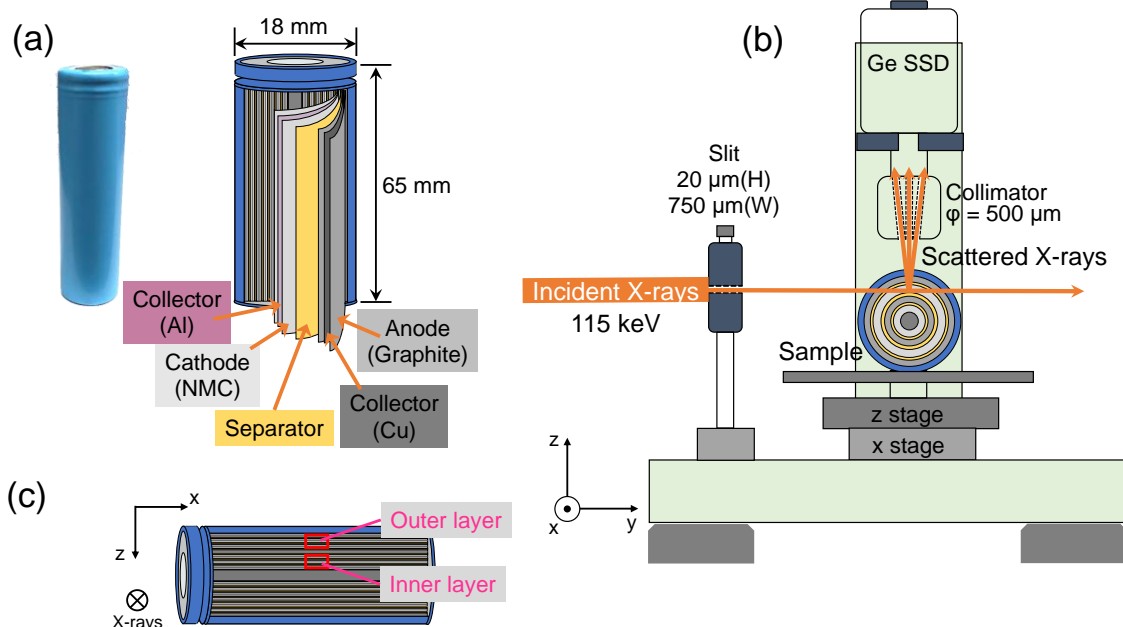

**Figure 1.** (**a**) A photograph and a schematic of the structure of the 18650-cylindrical cell. (**b**) Schematic view of Compton scattering imaging system at BL08W beamline of SPring-8. (**c**) Measurement regions: Compton scattering spectra were obtained from the two regions shown (outer and inner layers).

### 2.2. Compton Scattering Experiment and Data Analysis

Compton scattering experiments were performed at the high-energy X-ray beamline BL08W of SPring-8 (Sayo, Japan). The experimental setup is shown in Figure 1b. The energy of the incident monochromatic X-rays was 115.56 keV, and the scattering angle was fixed at 90°. The Compton scattered X-ray energy spectrum was measured by nine independent segments of a high-purity Ge solid-state detector (SSD). These nine detectors were adjusted to observe the same region of the sample. The Compton spectrum was obtained from two regions: one region near the outer casing of the cell (marked as outer layer in Figure 1c) and the other near the center of the cell (marked as inner layer in Figure 1c). We measured a region of about 1 mm in height and 0.75 mm in width by scanning incident X-rays. The spatial resolution of this measurement is controlled by the incident and collimator slits, which were 20 μm in height, 750 μm in width, and 500 μm in diameter.

The measured Compton scattered spectrum was converted to the line-shape parameter (*S*-parameter) defined as [9]

$$S = \frac{\int_{-d}^{d} J(p_z) dp_z}{\int_{-l}^{d} J(p_z) dp_z + \int_{d}^{l} J(p_z) dp_z} \tag{3}$$

where *d* and *l* are parameters that define low and high momentum regions, respectively. The Compton profile of lithium contributes to the low-momentum region [9,18], therefore, the *S*-parameter is related to the lithium concentration in the electrode. In particular, the *S*-parameter increases with increasing lithium concentration. In this study, we choose $d = 1$ atomic unit (a.u.) and $l = 5$ a.u. Details of post-processing for converting from Compton scattered energy spectrum to *S*-parameter are given in the Supplementary Materials.

### 3. Results and Discussion

In order to observe the inner structure of the fresh and aged cells, Compton scattered X-ray intensities were measured by scanning incident X-rays from the outer casing to the center of the cell following a procedure similar to that described by Suzuki et al. [9]. Figure 2 shows the corresponding results. Figure 2a,b shows the results of the fresh and aged cells, respectively. The open circuit voltage

(OCV) of the discharged and charged states of the fresh and aged cells was 2.628 V, 4.193 V, 2.677 V, and 4.125 V, respectively. The measurement of the discharged state was performed for 2 s per point, while that for the charged state was done for 1 s per point. The total number of points measured was 411 in each case.

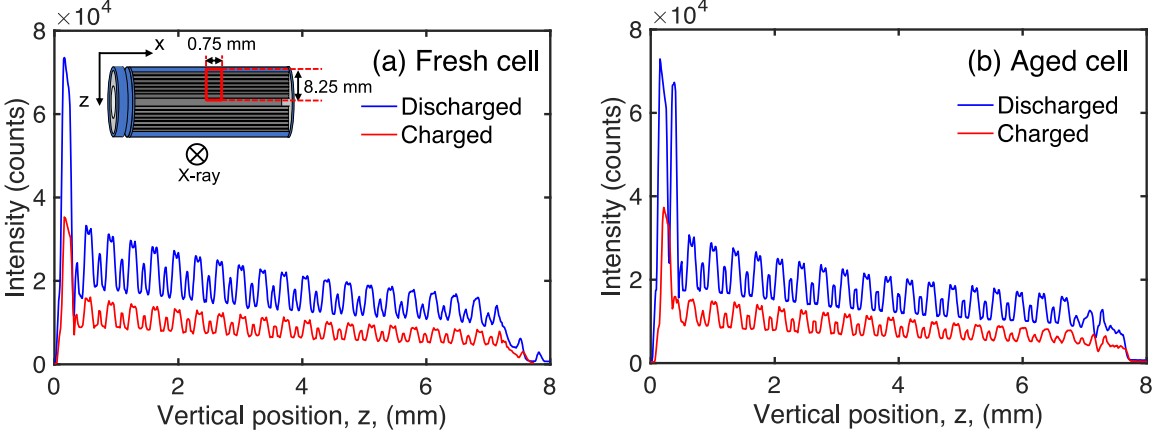

**Figure 2.** Structure of the fresh cell (**a**) and the aged cell (**b**) observed by Compton scattered X-ray intensities. Blue and red lines refer to discharged and charged states, respectively. The results were obtained by scanning incident X-rays at the region shown in the inset to (**a**).

This measurement was performed using a spatial resolution of 20 μm in height, 750 μm in width, and 500 μm in depth (note that a 20-μm height is small compared to the thickness of either electrode (0.19 mm for the anode and 0.15 mm for the cathode). Therefore, the layered structure can be clearly resolved as seen in Figure 2a,b in the fresh as well as the aged cells. In these graphs, the largest peaks around the vertical position of $z = 0$ mm correspond to the outer casing of the cell, the broad peaks correspond to the cathode and the Al collector, and the anode is located around the sharp peaks of the Cu collector. The decay of intensity in the inner layer reflects the absorption of X-rays in the matter. In order to focus on different lithiation states of the cells, the Compton scattered spectrum was measured in two selected regions: $0.35 < z < 1.3$ mm, which corresponds to the outer layer, and $5.6 < z < 6.6$ mm, which corresponds to the inner layer.

Figure 3 shows the values of the *S*-parameter extracted from two different regions of the fresh and aged cells by applying Equation 3 to the Compton scattered spectrum measured by scanning incident X-rays along the z-direction of the cell. Figure 3a,b shows the results for two different states of charging at the outer and inner layers of the fresh cell, while Figure 3c,d provides similar results for the aged cell. The background colors in Figure 3 identify the regions of the graphite anode (yellow), the NMC cathode (green), and the Cu/Al current collectors (gray). Notably, there is a separator between the cathode and the anode in our cell. However, we could not observe this separator because it is too thin compared with the incident X-ray beam size. The results in Figure 3a,b show that when the battery is charged, the *S*-parameter increases at the anode while it decreases at the cathode. This implies that lithium moves from the cathode to the anode when the cell is charged. The variation in the *S*-parameter between the charge and discharge in the aged cell is smaller than that in the fresh cell (Table 1). Here, the error bar for the *S*-parameter was estimated from the error in Compton scattered X-ray intensity in the low- and high-momentum regions based on Equation (3). The error bar for the averaged *S*-parameter was then determined by applying error propagation rules to the error bar of the *S*-parameter in each position. In the fresh cell, the average *S*-parameter of the outer layer decreases by about 1.9% at the cathode and increases by about 3.0% at the anode on charging. The average *S*-parameter of the inner layer decreases by about 3.3% at the cathode and increases by about 5.9% at the anode on charging. Thus, in the fresh cell, the values of the *S*-parameter show that a larger level of lithiation occurs in the inner layer. In contrast, in the aged cell, the average *S*-parameter of the outer

layer decreases by about 0.7% at the cathode and increases by about 1.1% at the anode on charging, while for the inner layer it decreases by about 1.1% at the cathode and decreases by about 0.5% at the anode. Since the change in the *S*-parameter corresponds to that in the lithium concentration, the *S*-parameter at the anode for the inner layer would be expected to increase in the aged cell, but the observed lithium concentration in the inner layer of the anode is not observed to vary significantly.

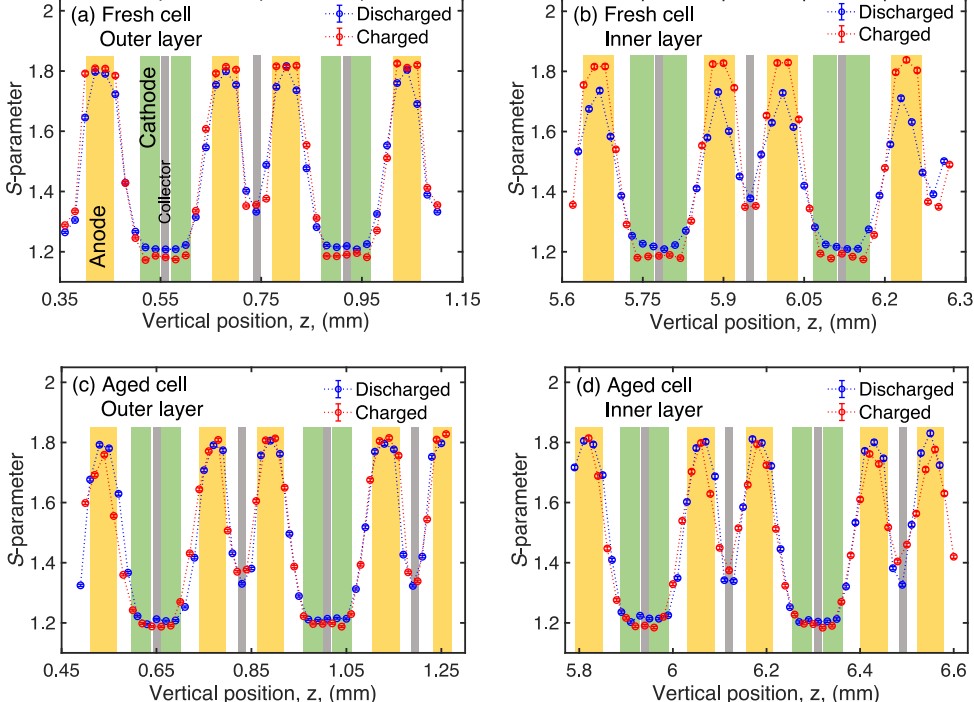

**Figure 3.** *S*-parameter obtained from the fresh and aged cells. Panels (**a,b**) show the *S*-parameter for the outer and inner layers in the fresh cell, respectively. Panels (**c,d**) show the *S*-parameter of the outer and inner layers in the aged cell, respectively. Blue and red dots indicate discharged and charged states, respectively. The background colors indicate the region of the NMC cathode (green), graphite anode (yellow), and Al/Cu current collectors (gray).

**Table 1.** Averaged *S*-parameters and changes between the charged and discharged states in the outer and inner layers for the fresh and aged cells.

|  |  | **Discharged** | **Charged** | **Relative Change** |
|---|---|---|---|---|
| **(a) Fresh cell Outer layer** | Cathode | 1.219 ± 0.001 | 1.196 ± 0.001 | 1.9 % |
|  | Anode | 1.756 ± 0.002 | 1.809 ± 0.002 | 3.0 % |
| **(b) Fresh cell Inner layer** | Cathode | 1.224 ± 0.001 | 1.184 ± 0.001 | 3.3 % |
|  | Anode | 1.709 ± 0.003 | 1.809 ± 0.002 | 5.9 % |
| **(c) Aged cell Outer layer** | Cathode | 1.211 ± 0.001 | 1.203 ± 0.001 | 0.7 % |
|  | Anode | 1.772 ± 0.002 | 1.791 ± 0.002 | 1.1 % |
| **(d) Aged cell Inner layer** | Cathode | 1.213 ± 0.001 | 1.200 ± 0.001 | 1.1 % |
|  | Anode | 1.788 ± 0.002 | 1.779 ± 0.003 | 0.5 % |

The *S*-parameters obtained for the fresh and aged cells are compared in Figure 4, where the upper and lower panels provide results for the outer and inner layers, respectively. Figure 4a,c refers to the discharged states whereas Figure 4b,d refers to the charged states. In the outer layer, the *S*-parameter obtained from either cell is found to show reproducibility in the discharged and charged states (Figure 4a,b). However, the *S*-parameter at the inner layer displays significant differences between

the two cells. Note that the *S*-parameters of the discharged state present large differences, as seen in Figure 4c. The most striking effect seen here is that the *S*-parameter of the anode does not appear to change even when the cell is discharged. Usually, the value of the *S*-parameter at the anode decreases, since lithium moves from the anode to the cathode during cell discharge. These results confirm that the degree of lithium mobility in the aged cell is smaller than that in the fresh cell. In connection with this, it has been reported that the formation of the solid electrolyte interface (SEI) on the surface of the anode provides a barrier for the motion of lithium [12,19].

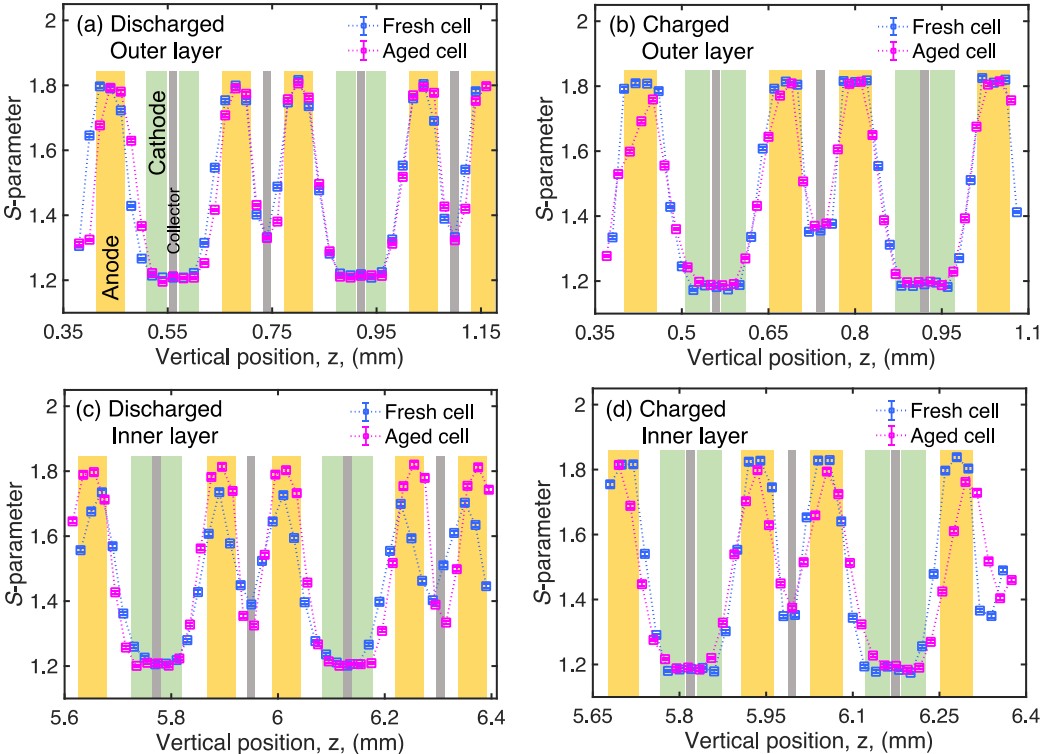

**Figure 4.** Comparison of the *S*-parameters for the fresh and aged cells. Upper figures are for (**a**) discharged state and (**b**) charged state at the outer layer, while the lower figures are for (**c**) discharged state and (**d**) charged state at the outer layer. Blue square dots are the data obtained from the fresh cell, while the pink square dots are data from the aged cell. The vertical position of the aged cell was shifted to fit the vertical position of the fresh cell.

## 4. Conclusions

The Compton scattering imaging technique was applied to examine the commercial 18650-type cylindrical cell for the first time. Using high-energy incident X-rays, we successfully measured lithiation states in the outer and inner layers of a large commercial cylindrical cell non-destructively. In the fresh cell, a significant lithiation change was observed in the inner layer. Moreover, by using the *S*-parameter analysis and comparing the fresh and aged cells, we adduced that the lithium mobility in the aged cell is smaller than that in the fresh cell. Our technique thus enables us to monitor the motion of lithium ions with micrometer-scale spatial resolution in operando in large cells, similar to our earlier results with coin type batteries. We note that measurements could be carried out more efficiently by using a pixelated detector instead of the Ge solid-state detector used in the present study. Moreover, as the next generation of synchrotron sources with higher energies and fluxes come online, it should become possible to monitor reactions in the electrolyte and analyze the lithiation state separately in each electrode with higher spatial resolution, and thus accelerate the development of high-performance lithium-ion batteries.

**Supplementary Materials:** The following are available online at http://www.mdpi.com/2410-3896/4/3/66/s1, Figure S1: (a) Cycling performance of the aged cell; (b) and (c) Impedance spectrum in the fresh and aged cells, respectively. Figure S2: Compton scattered X-ray energy spectrum obtained from nine Ge-SSD. Table S1: Areas under the low- and high-momentum region of the spectrum. Figure S3: Normalized Compton scattered X-ray energy spectra obtained from the charged and discharged NMC cathode in the fresh cell.

**Author Contributions:** Conceptualization, Y.S. and B.B.; experiment, K.S., A.-P.H., N.T., K.J., J.K., H.M., A.T., H.S., and Y.S.; formal analysis, K.S., D.H., H.H., H.S., and B.B.; resources, N.T., K.J., J.K., Y.S., M.K., S.H., A.B., H.S., and B.B.; data curation, all co-authors; writing—original draft preparation, K.S., H.H., H.S., and B.B.; writing—review and editing, all co-authors; supervision, Y.S., M.K., S.H., A.B., H.S., and B.B.; project administration, B.B.; funding acquisition, K.S., A.-P.H., S.H., and A.B.

**Funding:** The work at Gunma University was supported by the Association for the Advancement of Science & Technology (Gunma University) and Project for Functional Materials (Gunma University). The work at Northeastern University was supported by the US Department of Energy (DOE), Office of Science, Basic Energy Sciences (grant number DE-FG02-07ER46352), and benefited from Northeastern University's Advanced Scientific Computation Center (ASCC) and the NERSC supercomputing center through DOE (grant number DE-AC02-05CH11231). A.-P.H. was supported by University of Helsinki Doctoral Program in Materials Research and Nanosciences (MATRENA). A.-P.H. and S.H. were supported by the Academy of Finland (grant no. 1295696).

**Acknowledgments:** The Compton scattering experiment was performed with the approval of JASRI (Proposal Nos. 2018B1264 and 2019A1721).

**Conflicts of Interest:** The authors declare no conflict of interest.

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
