# Peer review of "High-Energy X-Ray Compton Scattering Imaging of 18650-Type Lithium-Ion Battery Cell"

_condensedmatter, doi:10.3390/condmat4030066_

Round 1

Reviewer 1 Report

This is a very good paper that shows how the lithium cell changes with ageing with Compton scattering imaging technique. The results are presented clearly and explained adequately. The paper is acceptable as it is written. One possible modification may be to change the wording on lines 76-77.

Perhaps it would be better to say that Compton scattering was applied to probe the structures instead of using the words "we propose" on lines 76-77. 

Author Response

Thank you very much for your constructive comments to our manuscript. We were happy to make appropriate revisions in the manuscript. The details are written in the replay to reviewer report and attached. Please find the attached file.

Reviewer 2 Report

This is a comprehensive study on the application of Compton scattering for measuring the state of lithiation locally inside commercial Li-ion cells. It is a well conducted experiment, is novel, and will be of interest to the battery and synchrotron community. Therefore, I recommend publication following some minor revisions.

There isn't much literature referenced in the introduction. It would be nice to see some more references to studies that look at local lithiation inside commercially relevant cells e.g. DOI: 10.1039/C8EE02373E      and       https://doi.org/10.1002/advs.201500332

Line 57: "quantitative analysis of lithium". Unless I misunderstood, this technique has not yet reached the sophistication to quantify the amount of lithium, but is currently able to tell whether there is more or less lithium present relative to a different point in time or space, which would be 'qualitative' rather than 'quantitative'.

Section 2.1: the electrode thicknesses of 0.19 mm and 0.15 mm seem quite high. Is this coating thickness of the electrode on the separator? If so, they are extremely thick electrodes for commercial batteries. I wasn't aware that any company was making such thick electrodes in 18650 cells. It's worth double checking.

Section 2.1: Can we see the degradation data of the aged cells? For example, by including some plots in supplementary material.

Section 2.2: I would like to see more information on the post-processing of the raw data. Does the detector consist of many pixels? If so, were the pixels readings averaged to generate the data in Figure 2? Also, what was the exposure time needed and could this be improved?

Section 3: What was the total number of points taken and how long did one complete depth-scan take? The battery community will be interested to know whether this is suitable for operando experiments.

Figure 3: In my version, I did not see any "green", "red" or "grey" regions in the figures. Perhaps the colors were forgotten or maybe there was an editing mistake? Please check. Without the colors I couldn't distinguish where the anode, cathode and collector materials began. Also, the "S" on the y-axis is overlapping with "parameter".

Can the authors speak more about how the errors in Table 1 were determined?

The authors conclusion that Li mobility is reduced in the aged cell is consistent with literature. Li get's tied up in the SEI layer and in lattice defects inside the active material.

In the conclusions section, I would like to see a short discussion on how this technique could be applied, perhaps, via operando experiments to elucidate lithiation heterogeneities inside commercially relevant cells. This would be of great interest to the battery community. I would also be interested in reading how the authors think this technique and experimental setup could be improved e.g. will synchrotron upgrades at Spring 8, ESRF and APS facilitate faster and higher resolution measurements?

Great work.

Author Response

(The authors gave the same response as above.)

Reviewer 3 Report

This manuscript focues on the hard X-ray high energy X-ray Compton scattering imaging for detecting the Lithium migration between two electrodes in fresh and aged commercial cells. With the advantage of the very deep penetration depth of 115 keV X-rays, the authors demonstrate the capability of this technique to probe a real-world battery cell with certain spatial resolution. I think this kind of technique will be valuable for battery characterizations and would very much like to recommend the publication.

I hope the authors could clarify some technical discussions so readers could understand better the work:

1.       Fig.2:  My understanding is that with the high-energy X-rays, the experimental signals are from everything covered in the route of the X-ray photons during the experiments, so the signals from different layers of electrodes/current collectors/separators all contribute to the signal. While I could understand the outer layers could be well resolved, when the probe moves towards inner layers, the signals of the outer layers should still contribute to the data, e.g., taking a penetration through the “curved” layers into the inner part of the 18650 cell (Fig. 1a) will cut through many layers from outer to inner layers, correct? If this is the case how the sharp peaks sustained all the way to almost 7.5 mm towards the center of the cell in Fig.2, and how the electrodes deep inside the cell could be well resolved for the further analysis in Fig.3/4?

2.       Please provide a brief explanation on how the S-parameter should be associated with the amount of Li in electrodes? This could justify better the validity of the analysis especially considering no change is observed in the aged cell as there is still decent amount of capacity left. 

3.       Please note that the “background colors … (red region)… (green region) … (gray region)” mentioned in the manuscript text (Line 151- line 153) seems to be missing in the Figure 3.

Again, I recommend the publication of this work and further review is not necessary.

Author Response

(The authors gave the same response as above.)

Round 2

Reviewer 2 Report

Authors have tackled all of my comments. I recommend publication of the updated manuscript.

Reviewer 3 Report

I am satisfied with the answers from the authors, and the additional explanations to the manuscript. I recommend the publication of this work as is.